

# Habitat suitability of cetaceans in the Gulf of Mexico using an ecological niche modeling approach

M. Rafael Ramírez-León[1], María C. García-Aguilar[2], Alfonsina E. Romo-Curiel[2], Zurisaday Ramírez-Mendoza[2], Arturo Fajardo-Yamamoto[2] and Oscar Sosa-Nishizaki[2]

[1] Posgrado en Ecología Marina, Centro de Investigación Científica y de Educación Superior de Ensenada, Baja California, Ensenada, Baja California, Mexico
[2] Departamento de Oceanografía Biológica, Centro de Investigación Científica y de Educación Superior de Ensenada, Baja California, Ensenada, Baja California, Mexico

Corresponding author
María C. García-Aguilar,
gaguilar@cicese.mx

## ABSTRACT

**Background**. The Gulf of Mexico (GOM) is a semi-enclosed sea where the waters of the United States, Mexico and Cuba converge. Al least 21 species of cetaceans inhabit it. The only mysticete (baleen whale) is found in the northeast (U.S. waters). The distribution of the 20 species of odontocetes (toothed cetaceans) is well understood in U.S. waters, but practically unknown in Mexican and Cuban waters. In this study we used sighting data from several odontocete species to construct habitat suitability maps in order to identify geographical regions suitable for high diversity throughout the GOM.

**Methods**. Historical datasets of georeferenced sightings from across the GOM were used to implement the maximum entropy algorithm (MaxEnt) to model the habitat suitability of each species. Five environmental predictors were used, selected for their influence over the occurrence of cetaceans: two oceanographic predictors (sea surface temperature and chlorophyll-*a* concentration), and three bathymetric predictors (depth, slope, and distance to 200-m isobath). A spatial approach based on the habitat suitability maps was used to identify the suitable regions.

**Results**. Only 12 species were modeled, which were the ones with the minimum sample size required. The models performed well, showing good discriminatory power and slight overfitting. Overall, depth, minimum sea surface temperature, and bottom slope were the most contributing predictor in the models. High suitability areas of 10 species were located on the continental slope, and four suitable regions were identified: (1) the Mississippi Canyon and the Louisiana-Texas slope in the northern GOM, (2) the west Florida slope in the east-northeastern GOM, (3) the Rio Grande slope in the west-northwestern GOM, and (4) the Tamaulipas-Veracruz slope in the west-southwestern GOM.

**Conclusions**. We were able to detect four geographic regions in the GOM where a high diversity of odontocetes is expected, all located on the continental slope. Although the methodology to identify them (spatial overlap) is a very conservative approach, it is useful for conservation and management purposes. The paucity of data did not allow all species to be modeled, which highlights the importance of establishing transboundary monitoring programs.

# INTRODUCTION

Understanding species' geographic distribution patterns and related environmental factors is a central topic of population ecology (*Guisan & Zimmermann, 2000*). Environmental factors include both abiotic conditions that influence the physiological response (e.g., temperature) determined in turn by the species' adaptive responses and the interspecific interactions (e.g., prey availability) (*Soberón & Peterson, 2005*; *Peterson et al., 2011*). Cetaceans are a group of fully aquatic mammals whose anatomical, morphological and physiological adaptations have allowed them to colonize a wide variety of aquatic habitats (*Katona & Whitehead, 1988*). Nevertheless, their distribution is usually explained in terms of the abundance of prey, primarily controlled by dynamic oceanographic conditions (e.g., sea surface temperature and mesoscale processes), as well as by physiographic features (e.g., bottoms depth and slope) (*Kenney et al., 1997*; *Forcada, 2018*).

The Gulf of Mexico (GOM) is a semi-enclosed sea connected to the Atlantic Ocean and Caribbean Sea. It has a 1.6 million km$^2$ surface and includes the Exclusive Economic Zones (EEZ) of Mexico (which represent ∼55% of the total surface of the GOM), the United States (∼40%), and Cuba (∼5%) (*De Lanza Espino & Gómez-Rojas, 2004*). There are at least 21 species of cetaceans that inhabit the GOM, including one mysticete, the Bryde's whale (*Balaenoptera edeni*), which is distributed exclusively in the northeastern GOM (*Soldevilla et al., 2017*), and 20 odontocetes (*Würsig, 2017*).

The distribution of cetaceans in the northern GOM (i.e., the U.S. EEZ) has been extensively studied. Based on sighting records, *Maze-Foley & Mullin (2006)* divided the cetaceans into two communities: (1) the continental shelf community, which includes the bottlenose dolphin (*Tursiops truncatus*), the Atlantic spotted dolphin (*Stenella frontalis*), and the Bryde's whale, and (2) the continental slope community, which comprises the remaining species, although the rough-toothed dolphin (*Steno bredanensis*) can be found in both. More recently, *Roberts et al. (2016)* used density surface models to describe the spatial distribution of cetaceans in the northern GOM. Overall, their results are consistent with those of *Maze-Foley & Mullin (2006)*, although they highlight the importance of the continental slope and submarine canyons, such as the Mississippi Canyon, as areas of high density of cetaceans. In contrast, the distribution of cetaceans in the southern GOM (i.e., the EEZs of Mexico and Cuba) is poorly understood. In fact, only one study has covered this region, but it was conducted by extrapolating data from the northern GOM (*Mannocci et al., 2017*).

Besides the ecological relevance of the GOM, it is an important economic area where fishing, tourism, and the hydrocarbon industry generate billions of dollars annually (*Karnauskas et al., 2013*), and it is a key transportation region (*Shepard et al., 2013*). Given its economic importance, the GOM ecosystem is under increasing anthropogenic pressure, threatening cetacean populations (*Roberts et al., 2016*). To determine the extent and impact of these hazards and to optimize threat mitigation and conservation measures, it is necessary to have accurate predictions of their distribution on a broader scale; that is, at the ecosystem

level. However, the latter is complicated given the limited data on cetaceans in the southern GOM (*Ramírez-León et al., 2020*), but one way to achieve this is to use ecological niche models (ENM).

ENM are statistical tools that define the distribution of suitable habitats of a species based on its ecological requirements (*Peterson et al., 2011*). The rationale is that the records, which are discontinuous in nature, are related to environmental and/or spatial characteristics (environmental predictors) to predict the suitable areas of the species in unsampled locations; therefore, the maps produced are spatially continuous, showing the regions where greater aggregation is expected (*Franklin, 2010*). Thus, areas of high habitat suitability are defined as those sites were ideal (or favorable) conditions exist for a species' long-term subsistence (*Peterson & Soberón, 2012*).

Our objective was to estimate the habitat suitability of the odontocetes of the GOM to identify those geographical regions that could support a high diversity of these cetaceans. The analysis included historical datasets of georeferenced sightings (presence-only data) recorded in both the south and north of the GOM. We used the maximum entropy (MaxEnt) modeling approach (*Phillips, Anderson & Schapire, 2006*), and five environmental predictors were selected based on their documented importance in determining the occurrence of cetaceans, either directly or indirectly: sea surface temperature, chlorophyll-*a* concentration, bottom depth and slope, and distance to the 200-m isobath (e.g., *Praca et al., 2009*; *Fernandez et al., 2018*; *Pace et al., 2018*).

## MATERIALS AND METHODS

### Study and modeling area

The modeling area was not restricted to the GOM, due to the high movement capacity of cetaceans, and because there are no physicals barriers in the marine environment for them. The area was extended to include the warm-temperate and tropical oceanic provinces of the northwest Atlantic Ocean (Fig. 1A; *Spalding et al., 2007*).

The physiography of the GOM is complex and consists of 13 physiographic sub-provinces (Fig. 1B). The continental shelf (≤200 m deep) can be very narrow, as the Tamaulipas-Veracruz shelf, or extensive, like the Yucatan and Florida shelves. The continental slope extends from the 200 m continental shelf break to 2,800 m depth, and there are vast canyons, such as the Mississippi Canyon. The oceanic zone extends beyond the slope up to the abyssal plain, where depths >3,500 m are reached (*Bouma & Roberts, 1990*; *Monreal-Gómez, Salas-de León & Velasco-Méndoza, 2004*). The GOM oceanic waters have oligotrophic conditions that contrast with the eutrophic coastal regions, which receive a high nutrient input by river discharges, mainly in the northern GOM (*Biggs, 1992*; *Lohrenz et al., 1999*; *Muller-Karger et al., 2015*).

### Presence-only data

Historical georeferenced sightings (presence-only data) of odontocetes were compiled. We discarded the Blainville's beaked whale (*Mesoplodon densirostris*), Gervais's beaked whale (*M. europaeus*), killer whale (*Orcinus orca*), and Fraser's dolphin (*Lagenodelphis hosei*) from our study because sightings of these species are infrequent (*Würsig, 2017*). The
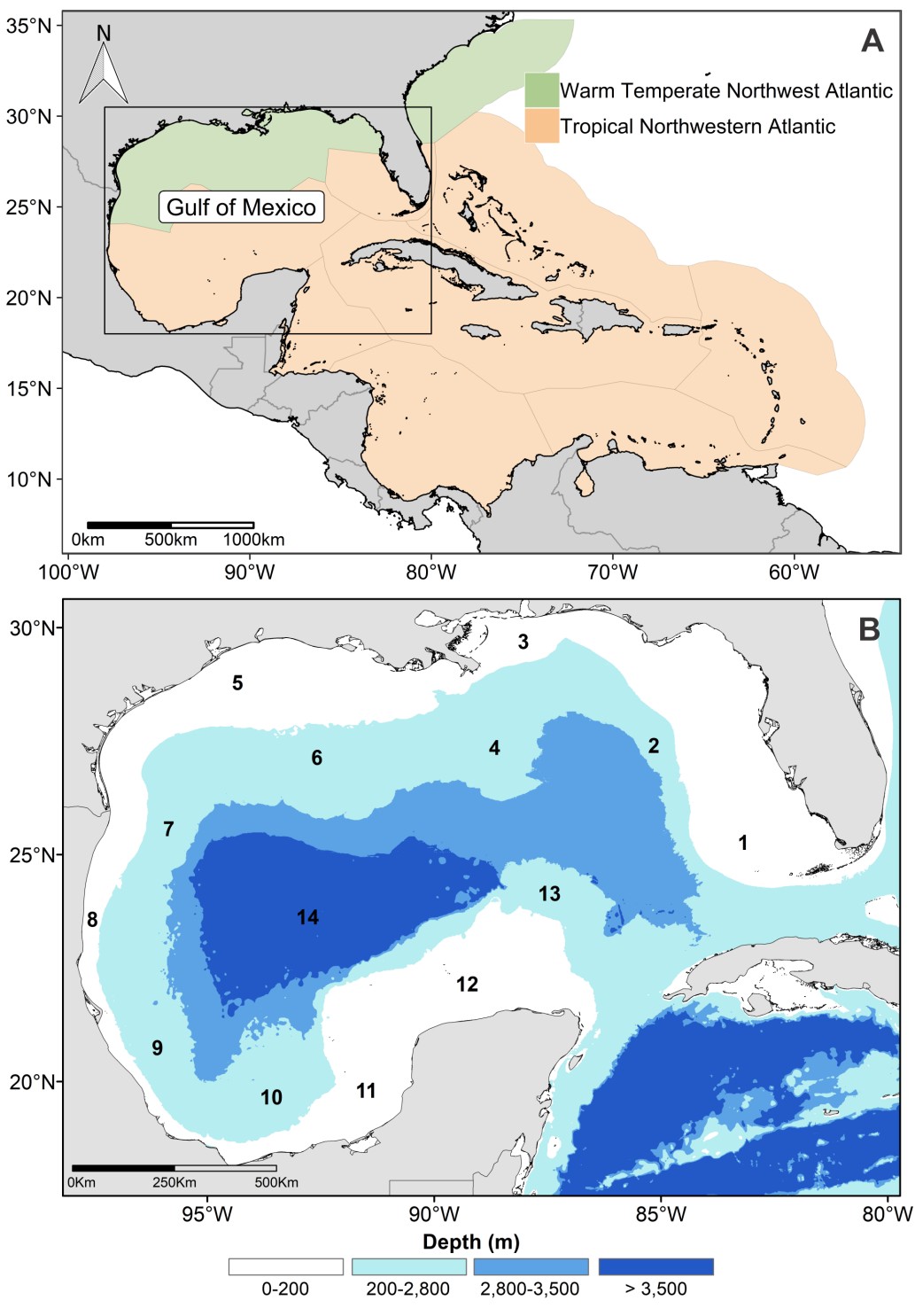

**Figure 1  Study and modeling area.** (A) Geographic extension of the modeling area in the northwestern Atlantic Ocean. (B) Gulf of Mexico and physiographic sub-provinces: 1. West Florida shelf, 2. West Florida slope, 3. Mississippi-Alabama shelf, 4. Mississippi Canyon, 5. Louisiana-Texas shelf, 6. Louisiana-Texas slope, 7. Rio Grande slope, 8. Tamaulipas-Veracruz shelf, 9. Tamaulipas-Veracruz slope, 10. Bay of Campeche. 11. Bank of Campeche, 12. Yucatan shelf, 13. Campeche terrace, and 14. Sigsbee plain.

**Table 1** **Georeferenced records for 16 odontocetes from the Gulf of Mexico.** Total number of georeferenced sightings of 16 odontocetes in the Gulf of Mexico, period of the presence data, filtering distance (average daily displacement in km), and number of sightings used in modeling (sample size). Modeled species are marked with an asterisk (*).

| Scientific name | Common name | Period | Total sightings | Filtering distance | Used sightings (n) |
|---|---|---|---|---|---|
| *Physeter macrocephalus** | Sperm whale | 1978–2017 | 810 | 90[1] | 70 |
| *Kogia breviceps* | Pygmy sperm whale | 1992–2011 | 51 | 75[2] | 17 |
| *Kogia sima** | Dwarf sperm whale | 1990–2011 | 319 | 75[2] | 37 |
| *Ziphius cavirostris** | Cuvier's beaked whale | 1990–2017 | 88 | 25[3] | 39 |
| *Feresa attenuata* | Pygmy killer whale | 1990–2008 | 24 | 70[4] | 16 |
| *Peponocephala electra* | Melon-headed whale | 1992–2011 | 70 | 70[5] | 25 |
| *Pseudorca crassidens* | False killer whale | 1986–2017 | 96 | 70[6] | 25 |
| *Globicephala macrorhynchus** | Short-finned pilot whale | 1984–2017 | 200 | 80[7] | 61 |
| *Steno bredanensis** | Rough-toothed dolphin | 1983–2017 | 90 | 90[8] | 37 |
| *Grampus griseus** | Risso's dolphin | 1990–2017 | 330 | 80[9] | 54 |
| *Stenella frontalis** | Atlantic spotted dolphin | 1979–2015 | 1,557 | 70[10] | 128 |
| *Stenella attenuata** | Pantropical spotted dolphin | 1983–2012 | 800 | 90[11] | 93 |
| *Stenella coeruleoalba** | Striped dolphin | 1992–2005 | 76 | 90[12] | 35 |
| *Stenella longirostris** | Spinner dolphin | 1983–2012 | 126 | 80[13] | 41 |
| *Stenella clymene** | Clymene dolphin | 1990–1998 | 108 | 70[12] | 37 |
| *Tursiops truncatus** | Bottlenose dolphin | 1971–2017 | 3,778 | 35[14] | 305 |

**Notes.**
[1] *Whitehead (2018)*
[2] *McAlpine (2018)*
[3] *Baird et al. (2009)*
[4] *Baird et al. (2011)*
[5] *Baird et al. (2012)*
[6] *Baird et al. (2010)*
[7] *Olson (2018)*
[8] *Wells et al. (2008)*
[9] *Wells et al. (2009)*
[10] *Davis et al. (1996)*
[11] *Scott & Chivers (2009)*
[12] *Gannier (1999)*
[13] *Perrin (2018)*
[14] *Irvine et al. (1981)*

presence-only data of the remaining 16 species (Table 1; Data S1) were collected from the literature (e.g., peer-reviewed articles, thesis, and technical reports), and digital databases of the Sistema Nacional de Información Sobre la Biodiversidad (SNIB; http://www.snib.mx/; *CONABIO, 2016*) and Ocean Biogeographic Information System Spatial Ecological Analysis of Megavertebrate Populations (OBIS-SEAMAP; http://seamap.env.duke.edu/; *Halpin et al., 2006*). To reduce the sampling bias (the north of the GOM is oversampled relative to the south) and the spatial autocorrelation that negatively affects model performance (*Boria et al., 2014*; *Varela et al., 2014*), we filtered our databases (one per species) using the spThin package (*Aiello-Lammens et al., 2015*) in R software (*R Core Team, 2019*). The *thin* function uses a random approach to return a dataset with the maximum number of records for a given distance restriction, which in this study it was defined by the average daily displacement of each species (Table 1; Supplemental Information 2).

## Environmental predictors

Five environmental predictors were selected based on previous knowledge about the environmental factors that influence the cetaceans' occurrence (e.g., *Praca et al., 2009*; *Fernandez et al., 2018*; *Pace et al., 2018*). The selected predictors included both oceanographic and bathymetric variables. Used oceanographic predictors were the sea surface temperature (SST, °C) and chlorophyll-*a* concentration (Chl-*a*, mg/m$^3$), included in three metrics: mean, minimum, and maximum. Data of both variables were downloaded from the Ocean Color portal (https://oceancolor.gsfc.nasa.gov/; *NASA, 2018*) of the MODIS-Aqua sensor for the period July 2002 –December 2018. The data are at an L3 processing level with a spatial resolution of 0.041° (∼4 km). Weekly values (8-d composite) were downloaded and averaged across the 16 years with available data. Bathymetric predictors were depth (*D*, m), bottom slope (*S*, degrees), and distance to the 200-m isobath ($D_{200}$, m). The first was acquired from the General Bathymetric Chart of the Ocean (GEBCO; https://www.gebco.net/; *IOCIHO, 2018*) with a spatial resolution of 0.008° (∼1 km); the other two were calculated from the depth using the raster package (*Hijmans et al., 2013*) in R software. These bathymetric predictors were re-projected at a spatial resolution of 0.041°.

The co-linearity among environmental predictors was evaluated using the Pearson correlation coefficient ($\rho$) (*Dormann et al., 2012*; *Cruz-Cárdenas et al., 2014*). If $\rho \geq 0.70$ (*Dormann et al., 2013*), a principal component analysis was performed (Supplemental Information 3) to determine which of the correlated predictors should be discarded.

## Habitat suitability modeling

We used the MaxEnt algorithm (*Phillips, Anderson & Schapire, 2006*) to predict the habitat suitability of odontocetes in the GOM. MaxEnt assumes that the species are distributed uniformly (i.e., the maximum entropy distribution) on the modeling area, and the environmental values constrain this distribution at the presence of records locations (*Phillips, Anderson & Schapire, 2006*; *Phillips et al., 2017*). The habitat suitability modeling for each species was conducted using the ENMeval package (*Muscarella et al., 2014*; *Muscarella et al., 2016*) in R. We built models with a random sample of 10,000 background points (i.e., points not registered as occurrence records in the modeling area that are contrasted with the occurrence positions) and select the *Linear, Quadratic,* and *Hinge* features of the MaxEnt algorithm. The cross-validation of the models was done using the block method that splits the presence data into four bins, three as training data and one as test data, based on the latitude and longitude lines that divided the occurrence localities (*Muscarella et al., 2014*).

The performance of each model was evaluated using the area under the receiver-operator curve (AUC), which measures the discriminatory ability of each model, and the omission rate (OR), which indicates the proportion of test localities that fall into cells not predicted as suitable (*Phillips, Anderson & Schapire, 2006*). An AUC of 1 indicates perfect discrimination between sites where the species is present or absent, and an AUC <0.5 indicates that the model performance is less than a random assumption (*Elith et al., 2006*). We used the 10-percentile training omission rate ($OR_{10}$) because it is less sensitive to the outlier presence

locations (*Radosavljevic & Anderson, 2014*). Omission rates greater than the expected value of 0.1 (or 10%) suggest model overfitting (*Peterson et al., 2011*; *Radosavljevic & Anderson, 2014*). Finally, we used the contribution percentages returned by each MaxEnt model to evaluate the contribution of each environmental predictor (*Phillips, Anderson & Schapire, 2006*).

We selected the logistic output and obtained the habitat suitability for each $0.041° \times 0.041°$ cell of the modeling area, which was expressed in an interval between 0 (unsuitable conditions) and 1 (highly suitable conditions). In this study, high suitability areas were defined as those sites (cells) with suitability values $\geq 0.6$ (*Kaschner et al., 2011*). A spatial approach based on the habitat suitability maps was used to identify suitable regions for cetaceans; that is, regions capable of supporting a high diversity of cetaceans (i.e., suitable regions) were defined as regions where the high suitability areas of at least seven species overlap.

# RESULTS

Habitat suitability was modeled for only twelve species (Table 1), which were those that after filtering had the minimum sample required ($\geq 30$ presence records; *Wisz et al., 2008*). The pygmy sperm whale (*Kogia breviceps*), pygmy killer whale (*Feresa attenuata*), false killer whale (*Pseudorca crassidens*), and melon-head whale (*Peponocephala electra*) were excluded due to small sample size (Table 1).

The models showed a good degree of discriminatory ability based on the AUC scores, which ranged from 0.74 (the pantropical spotted dolphin, *Stenella attenuata*, model) to 0.91 (the bottlenose dolphin model) (Table 2). On the other hand, the $OR_{10}$ value was close to the expected value in some models, such as the rough-toothed dolphin, but in others it was higher, as in the spinner dolphin (*Stenella longirostris*) model (Table 2), suggesting some degree of overfitting.

The environmental predictors used in each model differ (Table 2; Supplemental Information 4). The bottlenose dolphin models had the fewest predictors, while the Risso's dolphin (*Grampus griseus*), Atlantic spotted dolphin, pantropical spotted dolphin and spinner dolphin models had the most. Slope was included in 11 models, while both depth and minimum-SST in 10 (Table 2). However, in terms of contribution, depth was the most important environmental predictor, with a contribution of >25% in seven models, followed by the minimum-SST, which had an important contribution in five models.

High suitability areas for 10 species were located on the continental slope (Figs. 2–4). The pantropical spotted dolphin seems to be the species with the widest distribution, potentially occupying the entire continental slope. High suitability areas for the sperm whale (*Physeter macrocephalus*), short-finned pilot whale (*Globicephala macrorhynchus*), and spinner dolphin were found on the inner continental slope, while for the rough-toothed dolphin they were located both on the outer continental shelf and on the slope. Dwarf sperm whale (*Kogia sima*), Risso's dolphin, and Cuvier's beaked whale (*Ziphius cavirostris*) models show almost continuous high suitability along the northern continental slope, from Florida to Louisiana-Texas, even extending to the Rio Grande and Tamaulipas-Veracruz
**Table 2 Statistics of each model and contribution percentages of each environmental predictor.** Values of the area under the receiver operator curve (AUC) and of the 10-percentile training omission rate ($OR_{10}$), and percent of contribution of the environmental predictors in each model.

| Species | AUC | OR | Environmental predictors | | | | | | | | |
|---|---|---|---|---|---|---|---|---|---|---|---|
| | | | $SST_m$ | $SST_{min}$ | $SST_{max}$ | $Chl\text{-}a_m$ | $Chl\text{-}a_{min}$ | $Chl\text{-}a_{max}$ | $D$ | $S$ | $D_{200}$ |
| Sperm whale | 0.83 | 0.21 | | | 17.34 | | | 7.23 | 38.25 | 9.47 | 27.71 |
| Dwarf sperm whale | 0.86 | 0.28 | | 45.57 | 14.59 | | | 7.96 | 28.81 | 3.06 | |
| Cuvier's beaked whale | 0.85 | 0.24 | | 35.77 | 1.34 | | | | 41.51 | 4.92 | 16.46 |
| Short-finned pilot whale | 0.83 | 0.18 | | 15.25 | | 5.47 | | | 18.00 | 18.76 | 42.53 |
| Rough-toothed dolphin | 0.83 | 0.13 | | 57.93 | 1.10 | 25.45 | | | | 7.45 | 8.07 |
| Risso's dolphin | 0.87 | 0.27 | | 39.06 | 1.23 | 12.86 | | | 29.60 | 17.25 | |
| Atlantic spotted dolphin | 0.83 | 0.20 | | 6.95 | | 4.33 | | | 74.38 | 1.54 | 12.80 |
| Pantropical spotted dolphin | 0.74 | 0.25 | | 24.87 | 7.60 | 24.83 | | | 27.68 | 8.59 | 6.44 |
| Striped dolphin | 0.81 | 0.25 | 21.28 | | 13.91 | 37.04 | | | 16.85 | 10.91 | |
| Spinner dolphin | 0.80 | 0.38 | | 8.88 | 4.75 | 28.27 | | | 10.04 | 15.44 | 32.62 |
| Clymene dolphin | 0.88 | 0.16 | 44.88 | | 18.27 | | | 18.37 | | 14.44 | 4.03 |
| Bottlenose dolphin | 0.91 | 0.16 | | 20.18 | 1.07 | 2.91 | | | 75.84 | | |

**Notes.**

Environmental predictors: $SST_m$, mean sea surface temperature; $SST_{min}$, minimum sea surface temperature; $SST_{max}$, maximum sea surface temperature; $Chl\text{-}a_m$, mean chlorophyll-a concentration; $Chl\text{-}a_{min}$, minimum chlorophyll-a concentration; $Chl\text{-}a_{max}$, maximum chlorophyll-a concentration; $D$, depth; $S$, slope, $D_{200}$, distance to the 200-m isobath

slopes. High suitability areas for the Clymene dolphin (*Stenella clymene*) were found on the northern slope, with small patches on the Tamaulipas-Veracruz slope. Striped dolphin (*Stenella coeruleoalba*) model identified as high suitability areas some patches on the Tamaulipas-Veracruz slope and on the bay of Campeche. The Atlantic spotted dolphin model indicates that high suitability areas are located from the continental shelf to the inner slope of the entire GOM, whereas the model of the bottlenose dolphin points to the continental shelf, from Florida to the Tamaulipas-Veracruz.

The main region of high diversity of odontocetes was located between the Mississippi Canyon and the Louisiana-Texas slope (Fig. 5). Other suitable regions were identified on the west Florida slope and on the western continental slope, between the Rio Grande and Tamaulipas-Veracruz slopes.

## DISCUSSION

ENM are powerful tools for generating spatially explicit maps of species' habitat suitability. We used the MaxEnt approach to model the habitat suitability of the GOM odontocetes, using data from both the north and south, and managed to identify regions where high diversity can be expected. We decided to use this approach because it allows the development of reliable models of the potential distribution based on presence-only data, although it is important to emphasize that these models do not represent the probability of the presence of a species. However, due to the paucity of data, we were able to model only 12 of the 20 species present in the GOM. Furthermore, because we use historical sighting records, the resulting maps are integrated images that show no temporal variations, and no biotic interaction were considered. Biotic interactions might improve habitat suitability

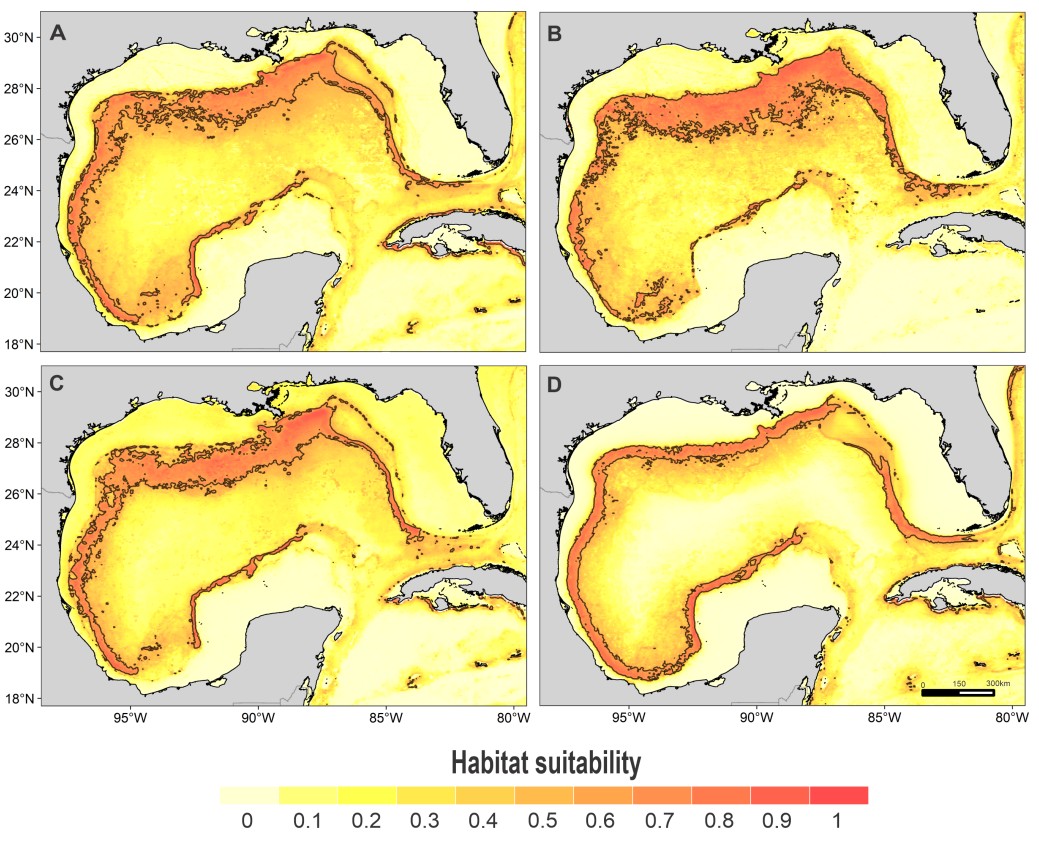

**Figure 2** **Habitat suitability map.** Habitat suitability of (A) sperm whale, (B) dwarf sperm whale, (C) Cuvier's beaked whale, and (D) short-finned pilot whale. In the scale bar, reds indicates high habitat suitability values (≥0.6) encompassed by the solid line, and light yellow indicates low habitat suitability values.

models, however, require abundance demographic data at population over time (*Anderson, 2017*).

All models had a good discrimination power, with AUC values >0.70, indicating that the results are reliable, and can be used in planning management and conservation measures (*Elith et al., 2006*; *Raes & Aguirre-Gutiérrez, 2018*). On the other hand, the $OR_{10}$ values were higher than the expected value; however, they are within the range reported in other studies (e.g., *Kramer-Schadt et al., 2013*; *Arthur, Morrison III & Morey, 2019*). High $OR_{10}$ values suggest overfitting, which could be due to sampling bias and/or noise in the presence data (*Anderson & Gonzalez, 2011*; *Merow et al., 2014*). In our study area, the south of the GOM is under-sampled compared to the north. We attempted to reduce this bias using the spatial filtering to minimize the omission error, but it may not have been completely successful, especially in the case of the spinner dolphin model.

Among the used environmental predictors, depth was the most important, followed by minimum SST and bottom slope. These results are not surprising since it is well known that these variables influence the occurrence of cetaceans directly; for example, some species display relatively persistent bathymetric associations (*Yen, Sydeman & Hyrenbach, 2004*;

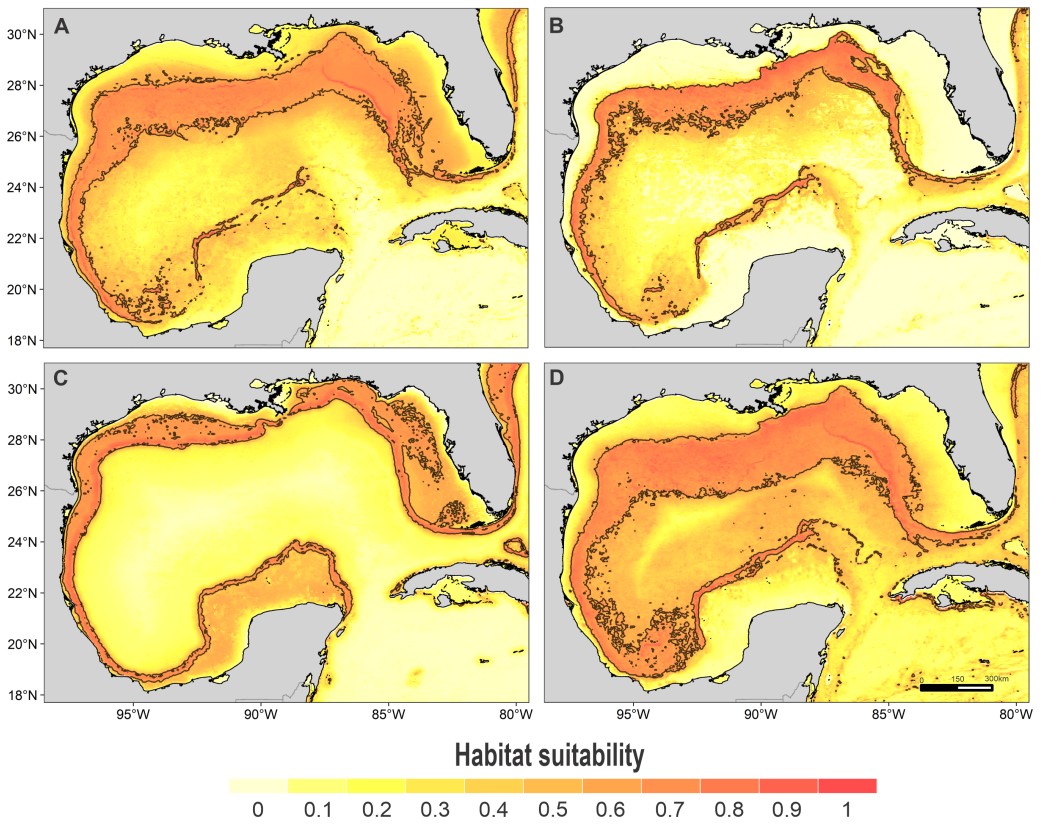

**Figure 3 Habitat suitability map.** Habitat suitability of (A) rough-toothed dolphin, (B) Risso's dolphin, (C) Atlantic spotted dolphin, and (D) pantropical spotted dolphin. In the scale bar, reds indicates high habitat suitability values ($\geq 0.6$) encompassed by the solid line, and light yellow indicates low habitat suitability values.

*Harvey et al., 2017*), but mainly indirectly by playing a determining role in the availability, distribution, and abundance of their prey (*Davis et al., 2002*; *MacLeod, 2009*; *Forcada, 2018*). In fact, previous studies have shown that the distribution of several species of cetaceans of the GOM is strongly related to depth (e.g., *Baumgartner, 1997*; *Davis et al., 1998*; *Baumgartner et al., 2001*).

Our results are consistent with the segregated distribution of cetaceans proposed by *Maze-Foley & Mullin (2006)* for the northern GOM. The two dolphin species of the shelf community, the Atlantic spotted dolphins and the bottlenose dolphins, use different habitats. High suitability areas of the Atlantic spotted dolphins were located on the outer continental shelf and the inner slope, while the bottlenose dolphin has coastal habitats, occupying shallower waters; actually, it is the only species that inhabits lagoons, estuaries, and bays (e.g., *Mullin et al., 1990*; *Griffin & Griffin, 2003*; *Martínez-Serrano et al., 2011*). The continental slope community is composed of the remaining species (*Maze-Foley & Mullin, 2006*; present study), although densities of these can vary seasonally, at least in the northern GOM (*Roberts et al., 2016*; *Mannocci et al., 2017*). The aggregation of multiple species reveals important biological regions capable of supporting a high cetacean

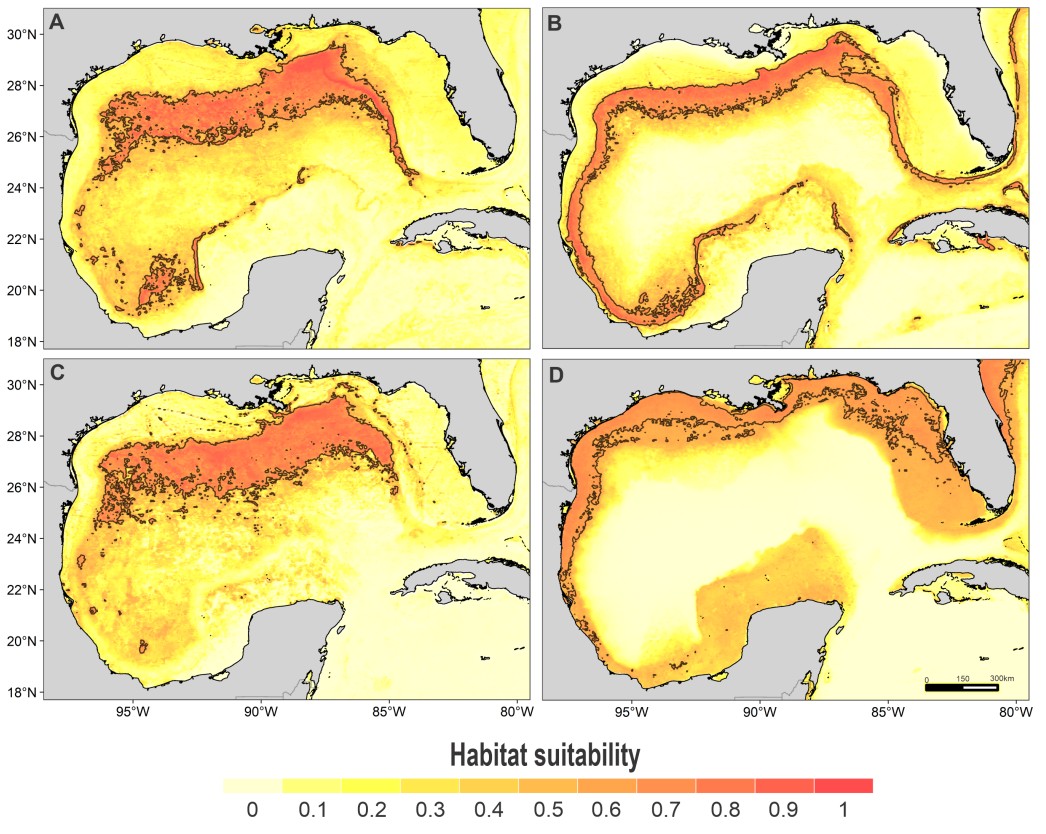

**Figure 4** **Habitat suitability map.** Habitat suitability of (A) striped dolphin, (B) spinner dolphin, (C) clymene dolphin, and (D) bottlenose dolphin. In the scale bar, reds indicates high habitat suitability values (≥0.6) encompassed by the solid line, and light yellow indicates low habitat suitability values.

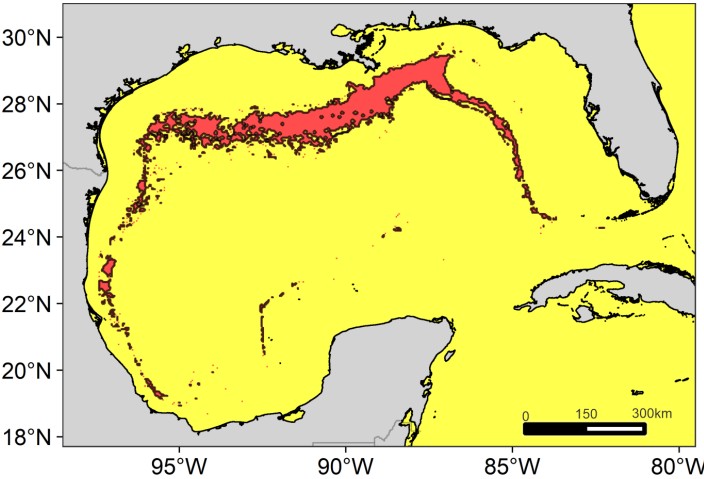

**Figure 5** **Suitable regions map.** Map of the overlap of the suitable habitat of cetaceans in the Gulf of Mexico. The solid line delimits the highly suitable regions with a high diversity of cetaceans ≥7 species.

diversity (*Harvey et al., 2017*). Ten of the 12 species modeled showed habitat suitability areas along the continental slope, consistent with the relatively high diversity of cetaceans observed on the northern GOM continental slope (*Davis et al., 2002*; *Maze-Foley & Mullin, 2006*; *Roberts et al., 2016*). The exploitation of different types of habitat and prey allows this co-existence (*Bearzi, 2005*; *Schick et al., 2011*). The continental slope of the GOM covers a large area and presents underwater canyons (*Bouma & Roberts, 1990*), where the main prey (e.g., cephalopods) of the deep-diving species can accumulate (*Biggs, Leben & Ortega-Ortiz, 2000*; *O'Hern & Biggs, 2009*; *Moors-Murphy, 2014*). On the other hand, the species that primarily feed on epipelagic prey preferentially use the upper layers of the water column, where mesoscale structures occur (*Davis et al., 1998*; *Davis et al., 2002*).

We identify four suitable regions. The most notable was located in the north, encompassing the Mississippi Canyon and the Louisiana-Texas slopes, consistent with that previously reported for the north of the GOM (e.g., Mullin & Fulling, 2006; (*Roberts et al., 2016*). The other regions were located on the west Florida slope (east-northeast of the GOM), the Rio Grande slope (west-northwestern of the GOM), and the Tamaulipas-Veracruz slope (west-southwestern of the GOM). All these regions are characterized by their high primary productivity. In the north, productivity is directly influenced by the input of nutrients from the Mississippi and Atchafalaya rivers (*Lohrenz et al., 1999*), while in the west by the Grande and Pánuco rivers (*Salmerón-García et al., 2011*). The plumes of nutrient-rich waters are transported through the continental shelf (*Del Castillo et al., 2001*; *Morey et al., 2003*; *Zavala-Hidalgo, Morey & O'Brien, 2003*), reaching the slope by the interactions of anticyclonic-cyclonic eddies (*Toner, 2003*; *Martínez-López & Zavala-Hidalgo, 2009*). However, the largest region suitable for cetaceans was located in the north for two possible reasons. First, the large nutrient input from the Mississippi River into the shelf ecosystem favors huge phytoplankton blooms on spatial scales of tens to hundreds of kilometers (*Lohrenz et al., 1997*). Second, the continental slope in this region is extremely wide, which could favor the convergence of a greater number of cetacean species.

## CONCLUSIONS

We identified areas of high suitability for 12 species of odontocetes in the GOM through the implementation of an ENM. Unfortunately, the paucity of data did not allow modeling all the species, which highlights the importance to establish transboundary research and monitoring programs between the U.S., Cuba, and Mexico to improve knowledge on the cetaceans of the GOM. Even so, we were able to detect four geographic regions where a high diversity of odontocetes is expected, all located on the continental slope. These suitable regions were identified using a spatial overlay, which although it is a very conservative approach (*Harvey et al., 2017*), it can be useful to detect areas where to focus conservation efforts (*Tolimieri et al., 2015*).

## ACKNOWLEDGEMENTS

This is a contribution of the Gulf of Mexico Research Consortium (CIGoM). We acknowledge PEMEX's specific request to the Hydrocarbon Fund to address the

environmental effects of oil spills in the Gulf of Mexico. We thank Paloma Ladrón de Guevara, Georgina Castro, and Fernanda Urrutia for helping to obtain and compile the occurrence data; Aimie Moulin and Rigel Zaragoza for helping to process the environmental layers, and Fabricio Villalobos and Carlos Yañez-Arenas for technical support; Sharon Herzka for key logistic support. We appreciate the suggestions of Ladd Irvine, Jason Roberts and one anonymous reviewer, which greatly improved the manuscript.

### Funding

This research has been funded by the Mexican National Council for Science and Technology—Mexican Ministry of Energy - Hydrocarbon Fund, project 201441. M. Rafael Ramírez-León held a Ph.D. scholarship from CONACYT. The funders had no role in study design, data collection and analysis, decision to publish, or preparation of the manuscript.

### Grant Disclosures

The following grant information was disclosed by the authors:
Mexican National Council for Science and Technology—Mexican Ministry of Energy—Hydrocarbon Fund: 201441.
CONACYT.

### Competing Interests

The authors declare there are no competing interests.

### Author Contributions

- M. Rafael Ramírez-León conceived and designed the experiments, performed the experiments, analyzed the data, prepared figures and/or tables, authored or reviewed drafts of the paper, and approved the final draft.
- María C. García-Aguilar conceived and designed the experiments, prepared figures and/or tables, authored or reviewed drafts of the paper, and approved the final draft.
- Alfonsina E. Romo-Curiel analyzed the data, prepared figures and/or tables, authored or reviewed drafts of the paper, and approved the final draft.
- Zurisaday Ramírez-Mendoza and Arturo Fajardo-Yamamoto analyzed the data, authored or reviewed drafts of the paper, and approved the final draft.
- Oscar Sosa-Nishizaki conceived and designed the experiments, authored or reviewed drafts of the paper, and approved the final draft.

### Data Availability

Raw presence data for each species of cetacean are available in the Supplemental Files.

### Supplemental Information

Supplemental information for this article can be found online at http://dx.doi.org/10.7717/peerj.10834#supplemental-information.

# PeerJ

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
