# Peer review of "Habitat suitability of cetaceans in the Gulf of Mexico using an ecological niche modeling approach"

_PeerJ, doi:10.7717/peerj.10834_

## Round 0.1 · original submission · Major Revisions

All three reviewers agree that this is an interesting study deserving publication. However, they have a number of comments that need to be addressed before the paper can be accepted. See their comments below; please note the error present in an input dataset that will require fixing (see review #3).

Please provide a detailed point-by-point reply to the reviewers' comments, ensuring that they are all properly addressed.

I look forward to receiving the revised manuscript.

Reviewer 1 ·

Basic reporting

no comment

Experimental design

no comment

Validity of the findings

no comment

Additional comments

This manuscript built maximum entropy (MaxEnt) models using oceanographic variables (sea surface temperature SST and chlorophyll-a concentration) and bathymetric variables (depth, slope, and distance to 200 m isobath) to predict the potential distributions of 12 cetacean species in the Gulf of Mexico (GoM), and to identify the most important environmental predictors influencing their distributions. The paper is interesting and potentially useful, as it demonstrated the importance of depth, bottom slope, and minimum SST in influencing the distributions of cetaceans in the GoM, and made an attempt to assess the habitat suitability of cetaceans along the continental slope of the GoM using MaxEnt. I think that the combined use of oceanographic and bathymetric predictors in a MaxEnt modeling framework is an interesting advantage of this paper in relation to others that deal with similar topics. There are some justifications the authors need to include, which will enrich the content of the research while clarifying the selection and implementation of the approaches used. The specific comments are:

Abstract
- (1) Lines 25. “MAXENT” to “MaxEnt”.

Introduction
- (2) Lines 49-54. This sentence seems too long. I would suggest rephrasing it into shorter sentences.
- (3) Line 52. “abundance of prey” to “the abundance of prey”.
- (4) Line 68. “can also be found in both” to “can be found in both”.
- (5) Line 84. “Presence-only data” to “presence-only data”.
- (6) I would suggest emphasizing somewhere in the Introduction section the significance of the combined use of oceanographic variables (SST and chlorophyll-a concentration) and bathymetric variables (depth, slope, and distance to 200 m isobath) in a MaxEnt modeling framework, because these approaches might help people from broader field find your work useful rather than just people working on cetaceans being interested in it.

Materials and Methods
- (7) Line 103 (and others throughout the manuscript). “Gulf” to “GoM” or “gulf”.
- (8) Line 113. “with ≥ 30; presence records” to “with ≥ 30 presence records”.
- (9) Lines 114-115. Are there any absence data of cetaceans in the GoM available from the literature or digital databases? There are some other modeling approaches (e.g., generalized additive models) that have been used in ecology when it comes to predicting species distributions. Given the potential of species distribution modeling to influence conservation or management, it may be necessary to critically assess the choice of modeling approach for the available data and specific environment considered.
- (10) Line 115. “peer review articles” to “peer-reviewed articles”.
- (11) Line 119. “(Supplemental Data S1)..” to “(Supplemental Data S1).”.
- (12) Line 119. I would suggest using “e.g.” rather than “i.e.”, because there might be other types of potential sampling bias.
- (13) Line 122. It should be “spThin package” rather than “spThing package”.
- (14) Line 123. What do you mean by “records that do not achieve the distance restriction of the nearest neighbor”? Do you mean records within the distance restriction or records outside the distance restriction? Please specify that.
- (15) Line 127. “each thinned datasets” to “each thinned dataset”.
- (16) Line 131. “factors that influence on the cetaceans’ distribution” to “factors that influence the cetaceans’ distribution”.
- (17) Lines 133-134. “Dynamic oceanographic predictors such as sea surface temperature (SST, °C), and chlorophyll-a concentration (Chl-a, mg/m^3) were included in three metrics” to “Dynamic oceanographic predictors, such as sea surface temperature (SST, °C) and chlorophyll-a concentration (Chl-a, mg/m^3), were included in three metrics”.
- (18) Line 135. When you examine the influence of SST and chlorophyll-a concentration on the potential distributions of cetaceans, I would suggest also considering seasonal/monthly variables and both concurrent and time-lagged (e.g., one season/month prior) data layers of SST and chlorophyll-a concentration, which may help better understand the distributions of cetaceans. Adding some discussion on the potential influence of time-lagged ecological variables in the Discussion section would also be helpful. In some cases, marine processes are better explained by time-lagged ecological variables, such as surface temperature, rather than simultaneous values (Olden and Neff 2001). Chlorophyll is often considered having a 30-day accumulation period prior to being reflected in higher trophic levels through ocean food chains (Trujillo and Thurman 2016; Wang et al. 2018).
Olden, J.D., and Neff, B.D. 2001. Cross-correlation bias in lag analysis of aquatic time series. Marine Biology 138:1063–1070.
Trujillo, A.P., and Thurman, H.V. 2016. Essentials of oceanography, 12th ed. (pp. 403–444), Boston, MA: Pearson Education, Inc.
Wang, L., Kerr, L.A., Record, N.R., Bridger, E., Tupper, B., Mills, K.E., Armstrong, E.M., and Pershing, A.J. 2018. Modeling marine pelagic fish species spatiotemporal distributions utilizing a maximum entropy approach. Fisheries Oceanography 27:571–586.
- (19) Line 140. “Static layers include three bathymetric predictors were considered” to “Static layers including three bathymetric predictors were considered”.
- (20) Line 144. It should be “raster package” rather than “Raster package”.
- (21) Lines 150-151. I would suggest including the results of principal component analysis (PCA) in the supplemental material, and adding a brief explanation of how you identified predictors that retained a high proportion of the original information based on PCA.
- (22) Lines 152-153. Did you use raw or log-transformed environmental data when applying the modeling approaches? It would be better to specify a little more even though you have removed the co-linearity among environmental predictors. Such data transformation may not be necessary for some situations, but it is an important step to make sure the variables meet the underlying assumptions of the algorithms before conducting any statistical analyses.
- (23) Lines 164-165. Some studies have attempted to use a distance-based function to restrict background points more likely to occur geographically further away from the presence records rather than randomly selecting them (e.g., Hengl et al. 2009). Have you ever considered that?
Hengl, T., Sierdsema, H., Radović, A., and Dilo, A. 2009. Spatial prediction of species’ distributions from occurrence-only records: combining point pattern analysis, ENFA and regression-kriging. Ecological Modelling 220:3499–3511.
- (24) Line 173. Did you use any cross validation when evaluating the model performance? I would suggest using cross validation when assessing the model performance. The predictive capability of a model could be overestimated without proper validation, because predictive power is generally highest on the training dataset, followed by approaches based on bootstrapped data resampling, and then on models validated using independent datasets (Wenger and Olden 2012).
Wenger, S.J., and Olden, J.D. 2012. Assessing transferability of ecological models: an underappreciated aspect of statistical validation. Methods in Ecology and Evolution 3:260–267.
- (25) Lines 177-178. The omission rate (OR) is typically threshold dependent. I would suggest specifying how you chose the threshold for this metric when assessing the model performance.
- (26) Line 182. Do you have any reference for why you chose 50% as the threshold of cumulative percent contribution?

Results
- (27) Line 191. I would suggest adding some explanations of why the sampling bias was persistent in the habitat suitability models of these species in the Discussion section.
- (28) Line 197. “models had the highest number” to “models had the most predictors”.
- (29) Line 202. “habitat-suitability” to “habitat suitability”.
- (30) Line 203. “high-suitability” to “high suitability”.
- (31) Line 224. “the entire Gulf, mainly on areas” to “the entire GoM, mainly in areas”.
- (32) Line 278. “Highly suitability areas” to “Highly suitable areas”.

Discussion
- (33) Line 291. “enriched-nutrients” to “nutrients-enriched”.
- (34) Line 295. “surface temperatures 29-30℃” to “surface temperatures of 29-30 ℃”.
- (35) Line 297. “in the east, these differences” to “in the east, and these differences”.
- (36) Line 301. “than in the eastern” to “than in the eastern region”.
- (37) Line 312. “composed by three species” to “composed of three species”.
- (38) Line 315. “have a different habitat” to “have different habitats”.
- (39) Lines 316 and 318. Do you mean “habits” or “habitat”?
- (40) Lines 322-323. “Both species do not show a dominant spatial pattern” to “Neither species shows a dominant spatial pattern”.
- (41) Line 330. “feed principally on” to “feed primarily on”.
- (42) Line 333. “the Mississippi canyon, and the west Florida and the Louisiana-Texas slopes” to “the Mississippi canyon, the west Florida and the Louisiana-Texas slopes”.
- (43) Line 361. “Presence-only records” to “presence-only records”.
- (44) Line 364. “Those our results” to “Our results”.
- (45) Line 375. “We identified the Mississippi canyon and the west Florida and the Louisiana-Texas slopes as” to “We identified the Mississippi canyon, the west Florida and the Louisiana-Texas slopes as”.
- (46) The potential distributions of cetaceans may also be affected by biotic interactions (e.g., predators). I would suggest acknowledging in the Discussion section these additional factors based on literature and their potential influence on the distributions of cetaceans in the GoM.

Tables and Figures
- (47) Table 2 Line 5. There is no “n” in the table. Do you mean “Sum” = number of models?
- (48) Table 2. I would suggest adding the “%” symbol to the percent contributions of environmental predictors in each MaxEnt model.
- (49) Figures 2-4. I would suggest specifying in the figure captions what the color range of the scale bar indicates (e.g., red indicates highest habitat suitability, and light yellow indicates lowest habitat suitability), because audience who are not in this field may not be familiar with what the habitat suitability of 0-1 stands for. A good figure or table caption should make the figure or table understandable without reference to the main text.

·

Basic reporting

The written language is generally ok but not good. There are a number of minor grammatical errors which may be due to English not being the primary language of the authors (based on the author affiliations). See Line 60, Line 107, lines 140 – 141, line 286, line 291, line 357. These errors generally don’t take away from the manuscript, however there are some places where it makes it difficult to understand what is being argued.

The introduction does an adequate job, however the case for a lack of distribution information for species outside of the northern Gulf of Mexico could be made more clearly. See general comments in relation to my suggestion for a slightly different focus for the manuscript which may require parts of the introduction to be revised.

In my opinion the manuscript would be significantly improved by the addition of two figures: First a figure showing the location of all sightings used as model input (distinguished by species) so the reader can get a sense of the spatial distribution of the sightings used and the biases that may be associated with them. Second, since the results are discussed as regions of the Gulf with highest habitat suitability across all studied species, a figure that presents the cumulative habitat suitability across all species should be presented so the reader can easily identify the high suitability areas rather than having to look at each figure and remember common regions across all figures. This could be something as simple as overlaying all the maps and summing or averaging the values of each grid cell.

Experimental design

The research question is explicitly stated however the justification could use some work. Since the justification is a lack of data outside the northern Gulf of Mexico, throughout the manuscript a better job could be done to distinguish studies/data from the northern Gulf of Mexico from the rest of the Gulf of Mexico/study results. I was often confused why study results were being presented as novel, along side data from the literature that seemed to show the same thing.

The methods used could use some additional detail as described in the specific line comments below. Biases associated with the data and model output should be discussed in more detail either in the Methods or in the Discussion. For example, sampling bias was mentioned on line 191 but no further detail was given about what it may have looked like and what its effects could be. A sample of 30 presence locations compared to 10000 available locations seems very small, so additional detail related to how Maxent performs on low-prevalence data and how model performance measures may be affected should also be added.

The lack of a temporal aspect to the environmental predictors used is also problematic. The modeling area is described as ‘dynamic’ but no effort to capture its variability was made. Further, some species being modelled may shift their ranges seasonally (as mentioned in the discussion). So the inclusion of parameters like min SST could actually be representing the suitability of an area for only a portion of the year, rather than for the entire year. These kind of issues should also be addressed in the section describing biases, etc, and ideally an explicit statement would be made about why the temporal aspect was not considered.

Validity of the findings

It doesn’t sound like model performance was tested on an independent data set. This is problematic and its implications should be discussed in the manuscript as you would expect the model to perform well on the data used to formulate it. If the input sightings did not represent all possible habitats of a species (likely given some species with relatively few sightings) the results may not be all that informative. With that said, if no other data truly exist for the non-US EEZ waters, any results may be a starting point to work from.

The interpretations of effects plots are strangely limited. Gradual increases in predictive value with decreasing min SST (Fig S2) are presented as essentially the same as strong increases in predictive value of min SST (Fig 3) by general statements like ‘… favored temperatures < 25 degrees’. I encourage the authors to give a little more consideration to what the results may mean.

Much of the discussion confuses this study’s correlative results with causation. Maxent only finds correlations in the data it is given and the actual process driving any observed trends is not addressed (and may not actually be observed). So results should be discussed based on their correlation with cetacean occurrence, rather than how SST/Depth/etc ‘drive’ cetacean occurrence.

Additional comments

General Comments:
The vast majority of marine mammal studies in the Gulf of Mexico have been limited to the US EEZ. This study sought to use a variety of sightings data to build habitat suitability maps for a wide range of cetacean species to define areas of suitable habitat throughout the Gulf of Mexico. This has the potential to be an interesting study, however, in my opinion, the way it is currently presented is challenging to understand and much of the methods need additional description or discussion of biases and related caveats so the reader can better understand how the results should be interpreted.

The discussion of areas of ‘high habitat suitability’ is confusing because the term seems to be used for both habitat suitability of individual species as well as areas of habitat suitability for multiple species. It seems like what is really being described are areas that are capable of supporting high cetacean biodiversity. Managing and conserving biodiversity is an area of emphasis throughout the world, so focusing on this terminology (identifying areas in the GoM that could support high cetactean biodiversity) will make the results more readily interpretable and of interest to a broader range of readers. The ‘suitable habitat’ terminology could then be used to describe individual species-level habitat associations, making a clear distinction between the two types of ‘high habitat suitability’ being discussed in this study. I personally think the identification of areas that could support high biodiversity is the most interesting aspect of this study but it is not covered very well. I would also suggest the authors consider focusing their language on ‘the distribution of suitable habitat’ rather than the currently used ‘potential distribution’ of a species as Maxent pretty explicitly states the output is not related to species density or probability of occurrence and in my opinion, nothing is gained by the use of potential distribution.

Line Comments:
Lines 22 – 24: This makes it sound like no SDM’s have been made for cetaceans in the GoM. Also, it doesn’t really make sense to mention 20 cetacean species if you are only looking at 12 of them
Lines 29 – 30: The most consistently significant predictors? Or supported? Constant isn’t quite the right word here
Lines 33 – 36: Were these ‘high-suitable regions’ identified that way because they were suitable for many species? I assume a separate model was fit for each species? So maybe they are highly suitable areas for cetacean biodiversity?
Lines 80 - 83: It is unclear if this is referring to high habitat suitability for each species or areas of high suitability for many species. Consider presenting the results in a biodiversity context where regions with high habitat suitability for many species would be considered regions capable of supporting high cetacean biodiversity. Then ‘habitat suitability’ could be reserved to discuss habitat of individual species. Also consider revising the wording related to geographical regions for clarity as it is often difficult to distinguish between what is known/being discussed about the northern Gulf/US EEZ and the rest of the GoM.
Lines 91 – 94: Why do the figures (except Fig 1) show just the GoM when the modelling area was the whole warm-temperate north west atlantic? Was this done to expand the number of sightings used as model input? If so it should be explicitly stated.
Lines 95 – 104: This would probably fit better in the introduction
Line 107: ‘As we mentioned’. It is unnecessary and can be deleted
Lines 112 – 113: I suggest starting the paragraph with this line (only selecting species with > 30 records) and then follow up with a justification for that threshold and what species that removed from the total.
Line 127: A map showing all sighting locations would be helpful to orient the reader and give the reader a sense of the possible spatial biases in the data
Line 129: Were any other predictors considered? Possibly Sea Surface Height or measures of vorticity? Especially with the more oceanic, deep-diving species, it seems unlikely that surface-oriented predictors will have much value.
Line 138: I’m not clear how an 8 day composite was used to characterize the annual cycle of each variable? The mean, min and max values over the 16 year time period were calculated on 8 day composites? Also, why were annual means used instead of the values at the time of each sighting? That would seem to be more relevant and would potentially help incorporate temporal aspects to distribution like seasonality.
Lines 150 – 153: How would a principal components analysis determine which correlated predictor should be removed? It seems like the mean, minimum and maximum of the predictors will always be strongly correlated?
Lines 159 – 160: This sentence needs some additional work as I’m not quite sure what is being said.
Line 167: What is ‘the block as data portioning’? Is this using cross validation methods? The sample size is awfully small to be holding out some of it during model development.
Lines 168 – 170: What output was chosen? Logistic? Cloglog?
Lines 173 – 179: Performance measures ideally need independent data for comparison. Was this used or are the measures testing the model performance on the same data used to formulate the model?
Lines 181 – 183: This sentence is confusing. Predictors were deemed not significant when the cumulative percent contribution of 50% was reached? Consider re-wording for clarity
Lines 191 – 193: The reader needs to know what this sampling bias is. At a minimum there should be a figure showing the location of all sightings and some additional context here would be useful (all species were observed during one expedition/in one area/etc)
Line 195: The methods state there were five environmental predictors, but here there are up to 6?
Line 211: This is minimum temperature right?
Line 218: Minimum temperature?
Lines 285 – 288: This study identifies environmental covariates that are correlated with sightings. This is not to be confused with a causative relationship as implied by this statement. We do not know if the predictors ‘influence the distribution of cetaceans’ only that values within a certain range are associated with sightings.
Lines 302 – 304: So why was no attempt made to account for seasonal changes in dynamic predictors?
Lines 333 – 337: How was this identified? Visually? Was there a numerical way of adding the habitat suitability scores across all species? It is important to state that these areas are high suitability for a large number of the species modelled. Just mentioning ‘high suitable regions’ could be talking about just one species. There should also be a section in the Results describing these areas of common high suitability across species as I would think this is the most interesting part of the paper.
Table 2 line 5: I do not see ‘n’ anywhere in the table. Is this perhaps ‘Sum’?
Table 2: ‘Species’ is missing an ‘s’. How is there such a high OR for Dwarf sperm whales when the AUC is so high?
Figure 4: It looks like vessel track lines can be seen in panel C (in the north, northwest section on the shelf) and can be seen faintly in panel A and B. What could cause that since the predictor variables were all remotely derived (not ship-based)?

·

Basic reporting

The article has the expected professional structure, with proper tables and figures, and sufficient references and background material. The raw data are provided, in the form of an Excel file containing the sightings used to fit the models. From this, the analysis could probably be reconstructed by obtaining covariates from the documented sources and fitting the models using the methodology described. The writing is generally good, aside from some minor wording issues (see below).

I find the Results section, where the species are each reported, to not be very interesting. Essentially these species subsections are just mechanical summaries of Table 2, the maps presented in Figures 1-4, and the response curves in Supplementary Information S2. Personally, I don’t find this dry narrative from lines 200-280 to be very helpful, because I can see all that information just by examining the plots and Table 2. But I recognize that it may be traditional to include such text in the Results section, so it is ok with me if you leave it in the paper. However, if you needed additional space, e.g. to expand the Discussion section, you could cut out lines 200-280, or further compress them. Also, you could draw polygons of high and moderate suitability on your maps, if you felt that communicating that information better would benefit your readers.

The paper would be improved by changing some words and correcting some minor grammar mistakes, enumerated below. However, it was easy to follow everything despite these minor issues.

Line 22: For “projecting” I suggest “the projection of” or “prediction of”.
Line 24: “module” should be “model”.
Line 28: You could probably remove the word “predictors”.
Line 29: “the most constant environmental predictors” sounds like they had the least variation. I suggest “the most frequently selected predictors”.
Line 33: Instead of “We identified in the northern GoM,” I suggest “In the northern GoM, we identified”.
Line 34: “high-suitable” should be “highly-suitable”.
Line 35: Throughout the manuscript, I suggest the term “moderately-suitable” rather than “medium-suitable”.
Lines 44-45: Because the word “species’” is plural, the word “response” should probably also be plural.
Line 51: “their distribution is” should probably be “their distributions are”.
Line 71: The common name for S. clymene is Clymene dolphin, with the C capitalized. Please capitalize the C of “Clymene dolphin” throughout the manuscript.
Line 80: “distribution” should probably be “distributions”.
Lines 82-83: The sentence starting with “Moreover” is not a complete sentence.
Line 84: “Presence-only” should not be capitalized. Also on line 361.
Lines 107-108: “GoM being the most of them” should be “GoM, most of them being”.
Line 119: Remove extra period.
Line 122: “spThing” should be “spThin”.
Line 127: “each thinned datasets” should be “each thinned dataset”.
Line 131: “on the cetacean’s distribution” should be “cetacean distributions”.
Line 132: “both the static” should be “both static”.
Line 134: I suggest “forms” instead of “metrics”.
Line 140: “include” should be “included”.
Line 141: Delete “were considered”; it is redundant.
Line 141: What does “grades” mean? I am not familiar with that unit.
Line 151: “need to” should be “needed to” or “should”.
Lines 152-153: I suggest deleting “therefore, only independent predictors were used”. Although collinearity is often mitigated using the technique you mentioned, and I don’t think you need to change your analysis, I’m not sure that the test of p >= 0.70 was sufficient to claim the predictors were “independent”. Therefore, I suggest you just delete that phrase.
Line 157: I suggest “from” should be “in”.
Line 160: “predictors” should be “predictor”.
Line 166: “selecting” should be “selected”.
Line 167: I do not understand “We applied the block as data partitioning”. Could you make this clearer, please?
Line 169: “and it was expressed” should be “which was expressed”.
Line 189: “0.75, according” should be “0.75; according”.
Lines 285-288: I do not understand what this sentence is trying to say. I suspect it is saying that although the two predictors do broadly influence the distributions of cetaceans in a direct way (e.g. through thermoregulatory constraints), they exert more influence in an indirect way by directly influencing the distribution of prey. Is that the right idea? Could you please rephrase the sentence to be clearer?
Line 303: The word “hypothesizing” does not make sense here. Who or what is doing the hypothesizing? In that sentence, the seasonal abundance changes are doing the hypothesizing. Please fix this.
Line 306: The word “Gulf” is capitalized here, but throughout most of the rest of the manuscript it is not. Please use consistent capitalization. (I do think “Gulf” is better than “gulf”, but either is ok.)
Line 308: “strategies, in turn, linked” should probably be “strategies, which are, in turn, linked”.
Line 313: What is “assed”? Please replace with the correct word.
Line 330: “use preferably” should be “preferentially use”.
Line 330: Remove the comma.
Line 355: I suggest “information” should be “knowledge” and “distribution” should be “distributions”.
Line 355: I suggest “to obtain” be changed to “obtaining”.
Line 357: I suggest this sentence be rephrased “As a result, robust knowledge of cetacean species distributions is sparse for several regions of the oceans, …”
Line 358: “because it allows to obtain” should be “because it facilitates development of”.
Line 363: Delete “prospectively;”. Also, I don’t understand what “ecological sound-based mitigation strategies” are. In particular, “sound-based” is odd.
Line 364: I believe “Those” should be “Thus”.

Experimental design

The article is within the aims and scope of the journal. The research question is well defined and the paper states how the research fills a knowledge gap (but see below). The investigation appears rigorous, and the methods appropriately selected, described, and executed well (but see below). Presence-only models were fitted for which there was strong potential for uneven sampling to bias the predictions, but the authors applied an appropriate mitigation to this common problem. To boost sighting counts and expand the range of sampling of model covariates, the analysis appropriately included sightings from outside the focal area but within the same ecoregions. Overall, the manuscript inspires confidence in the results. However, I have several concerns that should be addressed, numbered below.

1. Lines 69-77 of the Introduction review what is known of cetacean species distributions in the Gulf of Mexico, but two very relevant papers are missing. (Disclaimer: I am the lead author on one, and second author on the other.) Roberts et al. (2016, in Scientific Reports) produced density surface models for all species sighted by U.S. NOAA surveys of the Gulf of Mexico from 1992-2009. The spatial extent of these models was roughly the U.S. EEZ, comprising the northern ~40% of the Gulf, as mentioned on line 57 of the present paper. Then, Mannocci et al. (2017, in Conservation Biology) produced density surface models for most of the entire Gulf of Mexico and much of the western North Atlantic. Thus, the present paper is not correct when it states in lines 77-79: “However, so far, there are no accurate predictions of the distribution of cetaceans (i.e., spatially explicit maps) on a broader spatial scale spanning the entire GoM.” This statement should be revised to account for the fact that Mannocci et al. did provide such predictions, and the paper should mention (e.g. at line 77) both the Roberts and Mannocci models as prior attempts to model cetaceans in the GOM.

However, I do not believe the existence of Roberts’s and Mannocci’s prior results takes away from the value of the present paper, which may be distinguished from them in several ways. First, obviously, Roberts’s predictions only covered U.S. waters. Second, both of those papers involved density surface models, which give absolute density—the number of individuals per unit area—corrected for sampling effort. This is often very relevant for management applications, which often involve avoiding harm to animals where they currently reside. It might be argued that a density surface model depicts a realized niche, while a habitat suitability model developed from presence-only data, as in the paper under review, depicts a fundamental niche. However, this kind of thing has been heavily debated in the literature by theoretical ecologists, so if you endeavor to make this distinction in your paper, I suggest you carefully prepare proper citations. Finally, the present paper was able to utilize sightings not available for the Roberts and Mannocci analyses. Those analyses required line transect surveys compatible with distance sampling methodology (Buckland et al. 2001), while the present paper only required sightings compatible with presence-only modeling. That allowed the present paper to utilize additional data, particularly in Mexican waters of Gulf of Mexico, where Roberts and Mannocci did not have any data, and parts of the Caribbean.

This last difference is an important distinction. It makes the approach of the present paper potentially more suitable for modeling the southern half of the Gulf of Mexico. However, if you decide to discuss this, please be careful to review the advantages and disadvantages of both approaches. On the one hand, density surface models such as those by Roberts and Mannocci incorporate survey effort and therefore are much less likely to be biased than presence-only models. However, because no data suitable for that method was available for the southern Gulf, the models must be spatial extrapolated there, which is a risky procedure (for a review, see Yates et al. 2018, in Trends in Ecology and Evolution). On the other hand, the presence-only model under review here was able to incorporate data from the southern Gulf, so the degree of spatial extrapolation is greatly lessened. However, because presence-only models do not make use of effort data, risk is high that predictions may be inaccurate unless sampling bias is carefully addressed (see Fiedler et al. 2019, Frontiers in Marine Science, for a specific example concerning cetaceans). Finally, the two approaches model different things—absolute density (individuals per unit area) and habitat suitability (a unitless measure)—which are not equivalent.

Finally, the introductory sentence of the Discussion (line 283: “To our knowledge, this study constitutes the first approximation of the potential distribution of 12 resident cetaceans of the GoM.”) should be removed or altered, in light of the prior work of Mannocci et al.

2. Please include as supplementary information one map for each species showing the sightings enumerated in Table 1. Please show both the “total records” and the “used records”, with the used records layered on top and appearing with a different color or symbol, similar to Figure 1 of Aiello-Lammens et al. (2015) (but please use colors that provide more contrast than purple and red, which are hard to differentiate). This will allow readers to examine the spatial distributions of sightings and see the effect of the thinning procedure. This is very important, as thinning was the method used to mitigate sampling bias in this presence-only model.

3. Unfortunately, I discovered an error in one of the input datasets that affects two of the models. The analysis aggregated sightings from multiple datasets, as is often done with presence-only models. These included datasets from OBIS-SEAMAP, a bioinformatics repository for marine mammal sightings. One of the datasets, known as PIROP Northwest Atlantic 1965-1992 (http://seamap.env.duke.edu/dataset/280) and referred to as Hyrenbach et al. 2012 in the supplementary spreadsheet of sightings, had been processed incorrectly by OBIS-SEAMAP and accidentally switched species IDs for sightings of pantropical spotted dolphin (Stenella attenuata), of which very few were actually sighted, with Atlantic spotted dolphin (Stenella frontalis), for which many were sighted. I attached a PDF to this review showing the problem. This resulted in many sightings of S. attenuata appearing over the continental shelf of the eastern U.S. These should have been S. frontalis. I suspect this influenced the S. attenuata habitat suitability model to predict high suitability across much of the shelf of the northern Gulf of Mexico, even though almost no S. attenuata were ever sighted there (and those that were sighted were always close to the shelf break). This problem probably influenced the S. frontalis model as well, but those results do not appear to be as problematic, in my opinion.

Because of this problem, the pantropical spotted dolphin and Atlantic spotted dolphin models should either be fixed or be removed from the analysis. To fix them, you could either remove the Hyrenbach et al. (2012) dataset and refit the two models, or you could redownload this dataset from OBIS-SEAMAP, replace your current copy, and refit the models. The OBIS-SEAMAP data manager, Ei Fujioka (efujioka@duke.edu) has already fixed this problem after I reported it to him. He apologizes for this mistake. If you have any questions, you are welcome to contact him and myself (jason.roberts@duke.edu). The Results for these species should be updated accordingly, and Discussion at lines 315-323 adjusted as necessary.

4. I am unclear about whether all the predictors listed in Table 2 were actually used to make the final predictions. Lines 181-183 state “we identified the environmental predictors that best explain the data, we established a 50% threshold of the cumulative percent contribution (Supplemental Data S2).” Does this mean that you discarded less important predictors and based the final predictions on a model that included only the most important ones (that summed to >= 50%)? Or did you retain all the predictors shown in Table 2, but just use the 50% criterion to determine which predictors to discuss in the Results and to show in Supplemental Data S2? If it is the latter, which is suggested by lines 194-195, I think you should: 1) Adjust Supplemental Data S2 to include plots of all predictors. 2) Delete the sentences at lines 179-183. You do not need to explicitly state that this 50% criterion is what you used to decide which predictors to discuss. You do not need to change the text of the Results, in which you discuss only the predictors contributing >= 50%. It is ok to omit the less important predictors from your narration without mentioning why.

5. In the Supplementary Information S2, figures S9 and S11, the chlorophyll covariate is a flat line with slope of 0. This suggests that no matter what chlorophyll value occurred, the predicted value is the same. Is that indeed what is happening here? If so, please remove these terms from the model (because they had no influence on the predicted value) or provide some justification for keeping them in the model. Alternatively, these might be piecewise functions in which the predicted value is 0 outside of the flat line currently shown in the figures. If this is true, then please adjust the figures to show red lines at y=0 to indicate that the predicted value is 0 in those ranges.

Validity of the findings

Aside from the issues I noted in the “Experimental Design” section of my review, I believe the analysis is reasonable. The underlying data have been provided and were treated in a statistically sound way. Except for the two spotted dolphins I noted, I believe the results yield plausible habitat suitability maps. The Discussion section contains appropriate and interesting interpretations of the findings, with supporting citations. The Conclusions are well stated, with appropriate caveats.

Additional comments

Although I have written extensive comments, they are relatively easy to address and do not require any of the analysis to be changed, except for the two spotted dolphins. Because of that, I am recommending a “Major Revision” to the editor, because those results should be reviewed once they have been changed. I congratulate the authors on producing what I believe to be plausible habitat suitability predictions using a presence-only approach, which can be tricky to accomplish, given the potential for uneven sampling to bias the results. I hope that my comments have been helpful.

---

## Round 0.2 · Minor Revisions

Thank you for the revised version of your manuscript. All three reviewers and myself agree that the manuscript was considerably improved and their comments satisfactorily addressed. A couple of minor comments remain; they are appended below. Please address these by correcting the manuscript accordingly and/or providing a suitable reply.

Please also carefully check the text for minor grammatical errors and typos. Here are just a few I noticed:
l. 19 should be "where the waters of xxx converge"
l. 79 missing comma in reference (also l. 87)
l. 84 misplaced comma in reference
l. 171 should be "the block method that splits the data"
l. 236-237 word missing? maybe "the development of reliable models of the potential distribution"?

Reviewer 1 ·

Basic reporting

no comment

Experimental design

no comment

Validity of the findings

no comment

Additional comments

The authors have addressed most of the comments. I just have a few minor suggestions below, which I hope could help further improve the clarity of the paper.

- (1) For Response # 2, since you have some absence data (i.e., presence-absence data available for the north of the GoM, where cetacean surveys are carried out systematically), I would suggest trying another presence-absence modeling approach on the north of the GoM subset and comparing its performance with MaxEnt on the entire presence-only dataset. The comparison could be included in the supplemental material, and can be used to demonstrate if MaxEnt is still a better option when both presence and absence data are available.

- (2) For Response # 6, Line 181 (clear version), “Omission rates greater that the expected value of 0.1” to “Omission rates greater than the expected value of 0.1”.

- (3) For Response # 8, I still think it would be worthwhile to acknowledge in the Discussion section the potential influence of biotic interactions on cetaceans’ distributions in the study area based on literature. Sometimes physicochemical conditions may be an indirect reflection of biotic interactions, falsely suggesting a direct dependence on abiotic factors where in fact a biologically mediated dependence might be the case. The value of incorporating biotic interactions in improving the performance of habitat suitability models has been mentioned in quite a few studies and reviews (e.g., Van der Putten et al. 2010, Wang and Jackson 2014, Godsoe et al. 2015, Yates et al. 2018). Jointly modeling target species with their competitors, predators or facilitators has also been used as an option to achieve higher reliability and transferability of habitat suitability models (e.g., Zurell et al. 2009, Maguire et al. 2016).

References
Godsoe, W. et al. (2015) Information on biotic interactions improves transferability of distribution models. American Naturalist 185:281–290.
Maguire, K.C. et al. (2016) Controlled comparison of species and community-level models across novel climates and communities. Proceedings of the Royal Society B: Biological Sciences 283:20152817.
Van der Putten, W.H. et al. (2010) Predicting species distribution and abundance responses to climate change: why it is essential to include biotic interactions across trophic levels. Philosophical Transactions of the Royal Society B: Biological Sciences 365:2025–2034.
Wang, L. and Jackson, D.A. (2014) Shaping up model transferability and generality of species distribution modeling for predicting invasions: implications from a study on Bythotrephes longimanus. Biological Invasions 16:2079–2103.
Yates, K.L. et al. (2018) Outstanding challenges in the transferability of ecological models. Trends in Ecology and Evolution 33:790–802.
Zurell, D. et al. (2009) Static species distribution models in dynamically changing systems: how good can predictions really be? Ecography 32:733–744.

·

Basic reporting

The basic reporting is generally good although there are still a number of minor grammatical issues that could be resolved (for example Lines 65, 108, 180, 255, 265-266)

Experimental design

No comment, everything seems reasonable

Validity of the findings

Everything seems to be in order. I would suggest the authors consider adding a sentence or two to discuss why the suitability region in Figure 5 is so much larger in the northern Gulf compared to the western or southern Gulf.

Additional comments

The authors did a good job addressing and responding to my comments on the previous draft. I only have a few additional minor comments:
Lines 24 – 25: Maybe use language like ‘suitable for high diversity’?
Lines 86 – 88: Awkward wording
Lines 188 – 191: This could be simplified to something like “ ‘Regions capable of supporting a high diversity of cetaceans (suitable regions) were defined as regions where the high suitability areas of at least seven species overlap.’
Line 211: The figures should be referred to parenthetically after descriptive statements (like the second sentence) so this first sentence of the paragraph should be deleted.
Lines 249 – 250: Except you just stated that there was over-fitting. So maybe tone this down/revise it slightly to say you attempted to account for bias using the spatial filtering but it may not have been completely successful?
Figure 5: Additional detail in the caption is needed to make the figure understandable on its own. How were the areas of high diversity identified?

·

Basic reporting

In general, the manuscript has been substantially improved over the original submission. The authors largely addressed the points raised in my original review. The writing has also been improved and is largely free of grammatical errors. A few minor errors remain but they are not significant enough to block publication, and I'm not going to take the time to enumerate them here. However, if the editor does return the manuscript for additional revisions, I suggest the authors make one more pass through the text to check for proper English.

Experimental design

No comment

Validity of the findings

All habitat suitability maps now represent plausible results based on what has previously been reported in the literature. The model response curves appear to be generally consistent with what is known of the species for this region. The problem with the two spotted dolphin species I raised in my original review has been corrected.

Additional comments

I congratulate the authors on their improved manuscript and recommend it be accepted for publication.

---

## Round 0.3 · accepted · Accept

Thank you for this revised version and your replies to reviewers' comments. I am happy to approve this manuscript for publication.